# The Expression Patterns of Human Cancer-Testis Genes Are Induced through Epigenetic Drugs in Colon Cancer Cells

**DOI:** 10.3390/ph15111319

**Published:** 2022-10-24

**Authors:** Mikhlid H. Almutairi, Turki M. Alrubie, Bader O. Almutairi, Abdullah M. Alamri, Abdulwahed F. Alrefaei, Maha M. Arafah, Mohammad Alanazi, Abdelhabib Semlali

**Affiliations:** 1Zoology Department, College of Science, King Saud University, Riyadh 11451, Saudi Arabia; 442106519@student.ksu.edu.sa (T.M.A.); bomotairi@ksu.edu.sa (B.O.A.); afrefaei@ksu.edu.sa (A.F.A.); 2Genome Research Chair, Department of Biochemistry, College of Science, King Saud University, Riyadh 11451, Saudi Arabia; abdullah@ksu.edu.sa (A.M.A.); msanazi@ksu.edu.sa (M.A.); 3Pathology Department, College of Medicine, King Saud University, Riyadh 11451, Saudi Arabia; marafah@ksu.edu.sa; 4Groupe de Recherche en Écologie Buccale, Faculté de Médecine Dentaire, Université Laval, 2420 Rue de la Terrasse, Local 1758, Québec, QC G1V 0A6, Canada; abdelhabib.semlali@greb.ulaval.ca

**Keywords:** cancer-testis (CT) gene, DNA methyl-transferase inhibitor (DNMTi), histone deacetylase inhibitor (HDACi), immunotherapy, colon cancer

## Abstract

Background: The expression of human germline genes is restricted to the germ cells of the gonads, which produce sperm and eggs. The germline genes involved in testis development and potentially activated in cancer cells are known as cancer-testis (CT) genes. These genes are potential therapeutic targets and biomarkers, as well as drivers of the oncogenic process. CT genes can be reactivated by treatment with drugs that demethylate DNA. The majority of the existing literature on CT gene activation focuses on X-chromosome-produced CT genes. We tested the hypothesis that epigenetic landscape changes, such as DNA methylation, can alter several CT gene expression profiles in cancer and germ cells. Methods: Colon cancer (CC) cell lines were treated with the DNA methyltransferase inhibitor (DNMTi) 5-aza-2’-deoxycytidine, or with the histone deacetylase inhibitor (HDACi) trichostatin A (TSA). The effects of these epigenetic treatments on the transcriptional activation of previously published CT genes (*CTAG1A*, *SCP2D1*, *TKTL2*, *LYZL6*, *TEX33*, and *ACTRT1*) and testis-specific genes (*NUTM1*, *ASB17*, *ZSWIM2*, *ADAM2*, and *C10orf82*) were investigated. Results: We found that treatment of CC cell lines with 5-aza-2’-deoxycytidine or TSA correlated with activation of X-encoded CT genes and non-X-encoded CT genes in somatic (non-germline) cells. Conclusion: These findings confirm that a subset of CT genes can be regulated by hypomethylating drugs and subsequently provide a potential therapeutic target for cancer.

## 1. Introduction

Colon cancer (CC) is one of the top causes of mortality in the Kingdom of Saudi Arabia. CC typically affects older age groups; however, it has become more widespread among younger age groups in recent years [1]. It is the most prevalent cancer in men and the third most prevalent cancer in women in the Saudi population [2], and the highest incidence rates for men are found in Riyadh and Makkah [3]. Therefore, there is an urgent need for a reliable early identification method and a good therapeutic target for CC. Towards these goals, much research has been conducted on different classes of antigens to identify potential cancer-specific biomarkers and cancer treatments.

Cancer-testis (CT) antigens are examples of such molecules. In healthy adult tissues, CT antigens are only expressed in the germ cells of the testis; however, they are expressed in multiple types of cancers [2]. Therefore, CT antigens are extremely attractive immunotherapeutic targets due to their limited expression pattern and immunogenic character [2,4].

The expression of many CT antigen genes is controlled by the amount of methylation in the promotor regions of their DNA. When the amount of methylation is low, gene expression increases. Consequently, the expression of multiple CT antigens is upregulated in different cancers due to carcinogenesis-related genome-wide hypomethylation [4,5]. More importantly, the expression of CT genes can be increased in cancer cells by treatment with a DNA methyltransferase inhibitor (DNMTi); this is especially so in CT genes on the X chromosome [X-CT], where half of all known CT genes are located [4]. Almatrafi et al. (2014) showed that several CT genes are silenced by the hypermethylation of CpG islands and activated by DNA hypomethylating agents [4], and other studies have revealed a similar tendency for different CT genes [1,5,6]. For example, Colemon et al. (2020) observed a clear relationship between the methylation levels of MAGE-A genes and their expression in several cancer tissues [6]. Similarly, others observed that expression of the *NYESO-1* gene is increased by promoter hypomethylation in non-small cell lung cancer [7] and acute myeloid leukemia [8]. Hence, DNMTis have the potential to upregulate the expression of CT antigens and enhance the effect of immunotherapy [4,5]. For example, a DNMTi could be added to a clinical cancer immunotherapy treatment protocol.

CT gene expression is influenced not only by promoter methylation status but also by chromatin structure. Post-transcriptional changes that result in chromatin remodeling require histone acetyltransferases and histone deacetylases (HDAC). HDAC inhibitors (HDACis) appear to have anticancer activity and have entered clinical trials, indicating that they may contribute to a more individualized approach to cancer therapy [4,9].

A novel set of CT genes identified using in silico methods has been experimentally confirmed in cancer cell lines [10,11,12]. Recently, the expression of some of these genes was examined in samples from 20 Saudi patients with CC, and the results were compared to those from matched normal colon (NC) samples [2]. The genes *CTAG1A*, *SCP2D1*, *TKTL2*, *LYZL6*, *TEX33*, and *ACTRT1* were found to be expressed in the different CC tissue samples but not in any of the NC tissue samples, and most of these genes are located on autosomal chromosomes. This suggests that the six genes could be examined as potential CC markers. In addition, the genes *NUTM1*, *ASB17*, *ZSWIM2*, *ADAM2*, and *C10orf82* were not expressed in the CC and NC samples; therefore, they were categorized as testis-specific genes because they were only expressed in normal testis [2].

The hypothesis of this study was that epigenetic mechanisms, specifically DNA methylation, can regulate the expression of previously published CT genes or testis-specific genes in CC cell lines. In the current study, the effect of epigenetic treatments, namely the addition of a DNMTi and an HDACi, on the transcriptional activation of previously published CT genes was investigated. Specifically, CC cell lines were treated with the DNMTi 5-aza-2’-deoxycytidine (5-aza-CdR) or the HDACi trichostatin A (TSA). The results of this work further our understanding of the regulatory mechanisms of these genes in CC tissues.

## 2. Results

All CT genes that have been studied to date, which are mostly X-CT genes, require hypermethylation of their regulatory DNA sequences in order to be silenced. However, they can be activated by the hypomethylating agent 5-aza-CdR or/and by the HDACi TSA [4,10,11,12]. Given that the upregulation of immunogenic CT antigens might be useful in clinical settings, we aimed to determine whether recently found autosomal-encoded CT genes and testis-specific genes [3] were affected by a similar DNA hypermethylation silencing mechanism.

### 2.1. Effects of 5-aza-CdR and TSA on CT Gene Expression Profiles in CC Cell Lines

First, to investigate the effect of 5-aza-CdR on the expression of CT genes, the expression profiles of *SCP2D1*, *CTAG1A*, *LYZL6*, *TKTL2*, *ACTRT1*, and *TEX33* were examined. *CTAG1A* and *ACTRT1* are localized on the X chromosome, while the remaining genes are autosomes. Due to the dissolution of the 5-aza-CdR drug, DMSO-treated cells were utilized in this study. Furthermore, untreated cells served as negative controls. The normal testis served as a positive control for primer efficiency, while the *ACTB* gene was employed to assess the quality of the cDNA.

As controls for hypermethylation-regulated CT genes, two previously identified X-CT genes, *MAGE-A4* and *MAGE-B1*, were selected and almost remained transcriptionally silent in the Caco-2 and HCT116 cell lines. To examine whether a reduction in DNA methyltransferase activity can activate the selected CT genes, the two cell lines were treated with three doses of 5-aza-CdR (1, 5, and 10 μM) for 48 or 72 h. Following the 5-aza-CdR treatment, cDNA was synthesized, RT-PCR was performed, and the products were analyzed using agarose gel electrophoresis.

*MAGE-A4* and *MAGE-B1* were activated from a state of silence by a relatively low concentration of 5-aza-CdR, particularly in HCT116 cells. The novel X-CT *CTAG1A* gene was similarly activated with low levels of 5-aza-CdR in both cell types, while the expression of *ACTRT1* was not activated (1 µM; Figure 1). *SCP2D1*, a novel autosomal-encoded CT gene, was also activated only in Caco-2 cells with a slightly higher 5-aza-CdR dosage (5 µM; Figure 1). However, even at high concentrations of 5-aza-CdR in both cell lines, the transcription of *LYZL6*, *TKTL2*, and *TEX33* remained silenced (10 µM; Figure 1).

Next, qRT-PCR was performed to determine the levels of *MAGE-A4*, *MAGE-B1*, and *CTAG1A* mRNA in the Caco-2 and HCT116 cell lines following 72 h of treatment with 10 µM 5-aza-CdR. The expression level of each gene was measured in cells treated with 5-aza-CdR or DMSO. Then, the mean and standard deviation of the expression of each gene in each sample was calculated. The results indicate that the *MAGE-A4*, *MAGE-B1*, and *CTAG1A* mRNA levels were higher in cells treated with 5-aza-CdR than in cells treated with DMSO (Figure 2). Therefore, the qRT-PCR results correlate with the RT-PCR results, which are shown in Figure 1.

Second, Caco-2 and HCT116 cells were treated with 100 nM of TSA for 48 h to examine whether histone deacetylation was involved in the silencing of hypermethylation-independent genes (*LYZL6*, *TKTL2*, *ACTRT1*, and *TEX33*) and other novel CT genes. TSA was found to activate the expression of X-CT *MAGE-A4* in the Caco-2 and HCT116 cells and the expression of X-CT *MAGE-B1* in the HCT116 cells only. Surprisingly, *LYZL6* and *TEX33* remained firmly silenced under these transcription-inducing conditions (Figure 3). However, *TKTL2* transcription was activated only in HCT116 cells, while *ACTRT1* transcription was activated in both cell lines. In contrast, *SCP2D1* expression was induced in both cells; however, *CTAG1A* expression was stimulated only in the Caco-2 cells (Figure 3).

As a next step, qRT-PCR was performed to determine *MAGE-A4*, *MAGE-B1*, *CTAG1A*, *TKTL2*, *SCP2D1*, and *ACTRT1* mRNA levels in Caco-2 cells (Figure 4) and HCT116 cells (Figure 5) following 48 h of treatment with 100 nM of the TSA drug. The results indicate that *MAGE-A4*, *MAGE-B1*, *CTAG1A*, *SCP2D1*, *TKTL2*, and *ACTRT1* were more highly expressed in cells treated with TSA than in cells treated with DMSO (Figure 4 and Figure 5). Consequently, the qRT-PCR results correspond to the RT-PCR results, which are shown in Figure 3.

### 2.2. Effects of 5-aza-CdR and TSA on Testis-Specific Gene Expression Profiles in CC Cell Lines

In this study, the autosomal testis-specific genes *ADAM2*, *ASB17*, *C10orf82*, *ZSWIM2*, and *NUTM1* were validated to examine whether a reduction in DNA methyltransferase activity can activate these genes. Therefore, Caco-2 and HCT116 cells were treated with 5-aza-CdR. The expression of these genes remained silenced at all three doses of 5-aza-CdR (Figure 6). Additionally, to examine whether histone deacetylation was involved in the silencing of hypermethylation-independent genes, Caco-2 and HCT116 cells were treated with TSA, as described in Section 2.2. Remarkably, *ADAM2*, *ASB17*, *C10orf82*, *ZSWIM2*, and *NUTM1* remained firmly silenced under these transcription-inducing conditions (Figure 7).

### 2.3. Gene–Gene Interaction Network

The gene–gene interaction network of the CT genes was constructed to analyze the gene functions using the GeneMANIA database. The GeneMANIA online analysis tool showed that SCP2D1, CTAG1A, LYZL6, TKTL2, ACTRT1, and TEX33 are co-expressed with or are targets of 20 other genes: CTAG2, RGS11, PRAMEF2, MAGEA2B, PTH, GUCY2C, SPANXA1, MAGEA9, GLYATL1, TACR3, C1orf61, FAM133A, VHLL, PCDH15, HOXA2, FFAR1, MAGEA4, EDN3, TTN, and MYRFL (colored purple in Figure 8). The analysis also indicated that the relationship network of these genes was associated with co-expression. The central node representing the CT gene members was surrounded by 20 nodes, indicating genes that are strongly linked to the CT genes in terms of co-expression.

## 3. Discussion

The current treatment strategies for CC are restricted to surgery, radiation therapy, and chemotherapy, which are ineffective for patients with advanced recurrences and metastases, resulting in poor prognoses [13]. However, an immunotherapy that utilizes human immune cells to detect tumor-associated antigens is emerging as one of the most promising therapies for CC patients [14]. For cancer immunotherapy to be effective, it is necessary to identify an appropriate therapeutic target. Due to their high immunogenicity, restricted expression profile in normal somatic tissues, and widespread expression in different cancers, CT antigens are regarded as potential targets for cancer immunotherapy [15,16].

It has been shown that treating cancers with drugs that deregulate epigenetic silencing, such as those that reduce DNA methylation can result in the expression of CT antigen genes in cancer cells [17,18,19]. However, the epigenetic control mechanisms responsible for CT gene silencing have so far only been identified for a select few X-CT genes, all of which are triggered by hypomethylating drugs. In this study, we investigated the epigenetic control mediated by clinically significant indicators and demonstrated that a hypomethylating drug and an HDACi do not activate a subclass of CT genes that includes *LYZL6* and *TEX33*. These findings suggest that there is a wide range of mechanisms governing CT gene regulation. This has repercussions for the selection of CT genes for targeted therapies. Additionally, mechanistic regulatory pathways may reveal subclasses of CT genes that are co-regulated, which has significance for further research on these genes as biomarkers and/or therapeutic targets.

Furthermore, it has been revealed that some CT genes are essential for the growth of cancer cells. Inactivating these genes has the potential to reduce the impact of cancers and make other treatment techniques more effective by decreasing the proliferation-mediated burden of cancers. Interestingly, 5-aza-CdR treatment enhanced dose-dependent *CTAG1A* expression in the Caco-2 and HCT116 cell lines, suggesting that DNA hypomethylation is vital in the regulation of *CTAG1A* expression. These outcomes are consistent with those reported previously [7,8]. In contrast, the HDACi TSA triggered *CTAG1A* expression in Caco-2 cells but not in HCT116 cells at a similar dose, showing that not all cancers react to the same treatment.

## 4. Materials and Methods

### 4.1. Human CC Cell Line Sources and Cultures

HCT116 and Caco-2 human CC cell lines were used in this study. They were obtained from the American Type Culture Collection (ATCC; CCL-247 and HTB-37, respectively; Manassas, VA, USA). These cell lines were cultured with ATCC-recommended dilutions and confluences in DMEM (Thermo Fisher Scientific; 61965026) supplemented with 10% fetal bovine serum (Thermo Fisher Scientific; A3160801). They were incubated in a humidified environment with 5% CO_2_ at 37 °C.

### 4.2. Treating Cell Lines with 5-aza-2′ Deoxycytidine (5-aza-CdR) or Trichostatin A (TSA)

The HCT116 and Caco-2 cells were treated with three doses of 5-aza-CdR (Sigma; A3656) at 1, 5, or 10 μM for 48 or 72 h. Every 24 h, 5-aza-CdR-containing medium was added. The times and concentrations were selected based on the results of a previous study [4]. In addition, other sets of HCT116 and Caco-2 cells were treated with 100 nM of TSA (Sigma, T1952) for 48 h.

### 4.3. Total RNA Isolation and Complementary DNA (cDNA) Synthesis from Cell Cultures

According to the manufacturer’s recommendations, the total RNA was isolated from approximately 5 × 10^6^ cultured cells using the All Prep DNA/RNA Mini Kit (Qiagen, Hilden, Germany; 80204). A Nano-Drop 8000 spectrophotometer was used to determine the concentration, purity, and quality of the isolated RNA molecule (Thermo Fisher Scientific; ND-8000-GL, Waltham, MA, USA). For each sample, 2 μg of RNA was reverse transcribed into complementary DNA (cDNA) using a High-Capacity cDNA Reverse Transcription Kit (Applied Biosystems; 4368814), according to the manufacturer’s instructions. The cDNA was then diluted 10 times and stored at −20 °C until required.

### 4.4. Primer Design and Setup for Reverse Transcriptase PCR (RT-PCR)

All gene sequences were extracted from the National Center for Biotechnology Information database (available at http://www.ncbi.nlm.nih.gov/), accessed on 1 June 2022. For each gene, intron-spanning primers were designed to eliminate false positives that could be caused by genomic DNA contamination. Manual and software methods were used to select the primers. Primer-BLAST software was used to design the RT-PCR primers (available at https://www.ncbi.nlm.nih.gov/tools/primer-blast/), accessed on 1 June 2022. The primers were produced by Macrogen (Macrogen Inc., Seoul, South Korea) and were diluted in sterile distilled water to a final amount of 10 pmol (available at https://dna.macrogen.com/), accessed on 1 June 2022. The primer sequences for the RT-PCR genes are shown with their expected sizes in Table 1.

In a PCR tube, 10.5 μL of distilled water was combined with 12.5 μL of BioMix Red (BioLine, London, UK; BIO-25006), 1 μL of cDNA (100 ng), 0.5 μL of forward primer (10 pmol), and 0.5 μL of reverse primer (10 pmol). Each sample was amplified via a pre-denaturation step (5 min at 96 °C), followed by 40 cycles of a denaturation step (30 s at 96 °C), an annealing step (30 s at the temperature shown in Table 1, and an extension step (30 s at 72 °C). This was followed by a final extension step (5 min at 72 °C).

### 4.5. Agarose Gel Electrophoresis

The PCR products were run on 1% agarose gels (Sigma-Aldrich, St. Louis, MO, USA; A9539), using a 1× TBE buffer, and stained with 0.5 g/mL ethidium bromide (Sigma-Aldrich; 46067). The quality of the treated and untreated cDNA samples was determined using *ACTB* gene amplification. The size of each PCR product was determined by loading 3 μL of 100 bp DNA markers (New England BioLabs, Ipswich, MA, USA; N0467).

### 4.6. Primer Design and Setup for Real Time Quantitative PCR (qRT-PCR)

For efficient qRT-PCR amplification, all primers were manually designed with amplicon sizes of about 180 bp. To prevent internal secondary structure formation, each primer was 20 nucleotides long and included 50–55% G/C. To prevent primer dimer formation, the forward and reverse primers had equal melting temperatures and lacked considerable complementarity at their 3’ ends. To confirm the specificity of the primers, a BLAST search was conducted. Stock primers were diluted with sterile distilled water to 10 pmol. The sequences of the commercially made primers are displayed in Table 2.

The qRT-PCR tests were established with the manufacturer-recommended iTaq Universal SYBR Green Supermix (Bio-Rad, Hercules, CA, USA; 1725120). The volume was then adjusted to 10 μL by adding 5 µL of SYBR Green Supermix, 2 μL of cDNA (200 ng), 0.25 μL of each primer, and 2.5 μL of distilled water to a 96-well plate. Samples were amplified three times using a pre-denaturation step (30 s at 95 °C), followed by 40 cycles of 15 s at 95 °C, 30 s of primer annealing at 60 °C, and 15 s of extension at 95 °C. A melting curve study was conducted after the 40 cycles were completed. *GAPDH* was employed as a positive control for the normalization of the qRT-PCR data. Each qRT-PCR was conducted using a QuantStudioTM 7 Flex Real-Time PCR System (Applied Biosystems, Hercules, CA, USA).

### 4.7. Statistical Analysis

The qRT-PCR experiments were performed independently, two for each gene. The mean values and standard deviations were computed. GraphPad Prism version 5 (GraphPad Software Inc., San Diego, CA, USA) was used to identify significant differences in the outcomes. All *p* values were considered to be statistically significant (* *p* ≤ 0.05, ** *p* ≤ 0.01, *** *p* ≤ 0.001).

### 4.8. In Silico Analysis

A genetic interaction network of the CT genes and the functional associations was created using GeneMANIA tools (University of Toronto, Toronto, Canada). This was used to conduct a network analysis of common genes and to predict related genes [20].

## 5. Conclusions

The findings of this study demonstrate that the expression of some CT genes can be activated in CC cell lines by 5-aza-CdR and TSA agents. These epigenetic modulators play a crucial role in the transcriptional activation of these genes and represent possible components of future cancer immunotherapies. Additionally, the use of 5-aza-CdR and TSA in cancer vaccination therapies can be explored to improve the efficacy of this treatment modality. Additional research is necessary to determine the effects of 5-aza-CdR on protein levels at higher dosages when applied for longer durations and in combination with an HDACi.

## Figures and Tables

**Figure 1 pharmaceuticals-15-01319-f001:**
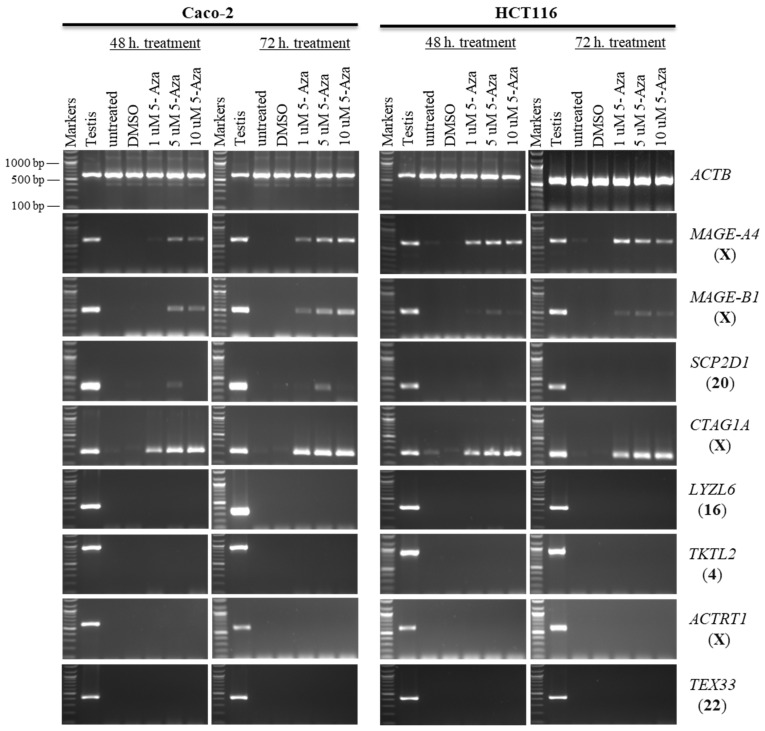
The effects of 5-aza-CdR treatments on CT gene expression profiles in the Caco-2 and HCT116 cancer cell lines. The expression of *SCP2D1*, *CTAG1A*, *LYZL6*, *TKTL2*, *ACTRT1*, and *TEX33* genes is shown on agarose gels following treatment with various doses of 5-aza-CdR for 48 h (left column of each cell) or 72 h (right column of each cell). Untreated Caco-2 and HCT116 cells were utilized as controls to compare the expression of CT genes with treated cells, and a testis sample served as a positive control for primer efficiency. The control Caco-2 and HCT116 cells were treated with DMSO, as DMSO was the solvent used in the 5-aza-CdR solution. The positive control for the cDNA samples is the expression of the *ACTB* gene. The official names of the genes are written to the right of the agarose gel images, and the location of each gene on the chromosomal is written in parentheses on the right. Above each lane, the particular concentration of 5-aza-CdR (1, 5, and 10 µM) is written.

**Figure 2 pharmaceuticals-15-01319-f002:**
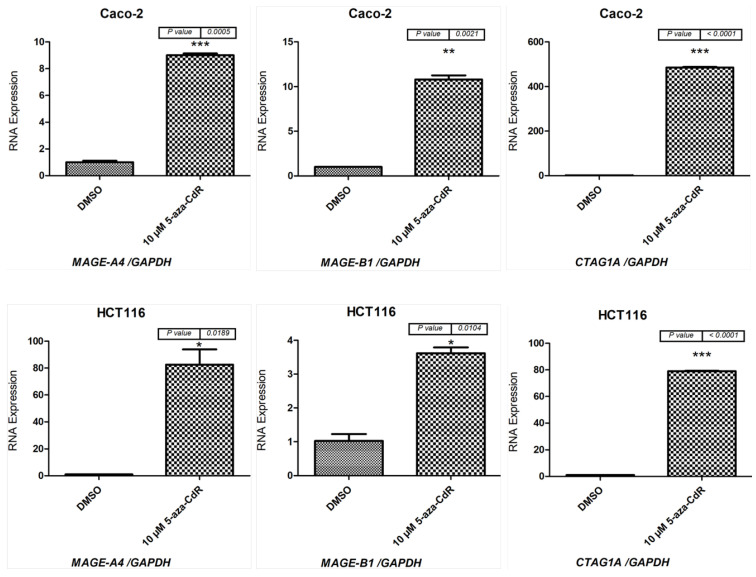
qRT-PCR analysis of *MAGE-A4*, *MAGE-B1*, and *CTAG1A* expression in Caco-2 and HCT116 cells following treatment with 10 µM 5-aza-CdR for 72 h. The bar charts illustrate the expression of *MAGE-B1*, *MAGE-A4*, and *CTAG1A* before and after 5-aza-CdR treatment in Caco-2 cells (upper panel) and HCT116 cells (lower panel). The expression levels were normalized to *GAPDH* mRNA levels. The error bars represent the standard error of the mean for three replicates for each gene. * *p* ≤ 0.05, ** *p* ≤ 0.01, *** *p* ≤ 0.001.

**Figure 3 pharmaceuticals-15-01319-f003:**
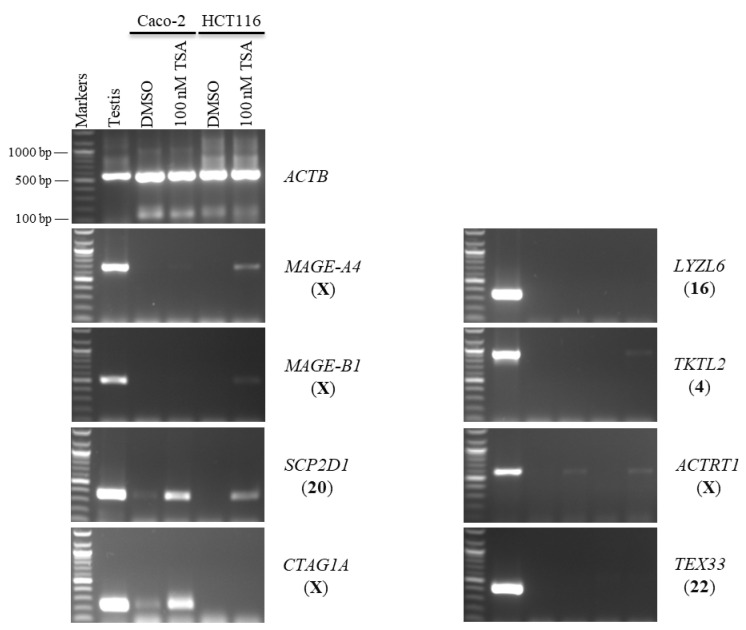
The effects of TSA treatments on CT gene expression profiles in the Caco-2 and HCT116 cancer cell lines. The expression of *SCP2D1*, *CTAG1A*, *LYZL6*, *TKTL2*, *ACTRT1*, and *TEX33* genes is shown on agarose gels following treatment with 100 nM of TSA for 48 h. The testis sample served as a positive control for primer efficiency. Control Caco-2 and HCT116 cells were treated with DMSO, as DMSO was the solvent utilized in the TSA solution. The positive control for the cDNA samples is the expression of the *ACTB* gene. The official names of the genes are written to the right of the agarose gel images, and the location of each gene on the chromosomal is written in parentheses on the right.

**Figure 4 pharmaceuticals-15-01319-f004:**
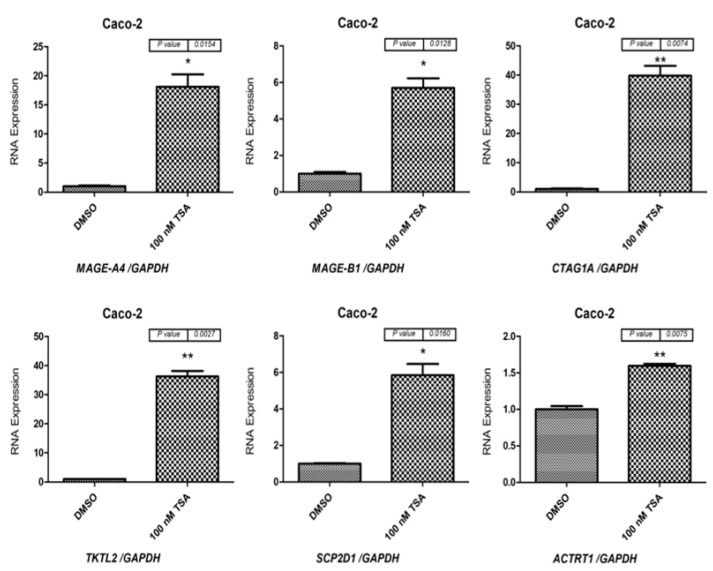
qRT-PCR analysis of *MAGE-A4*, *MAGE-B1*, *CTAG1A*, *TKTL2*, *SCP2D1*, and *ACTRT1* expression in Caco-2 cells following treatment with 100 nM TSA for 48 h. The bar charts illustrate the expression of *MAGE-A4*, *MAGE-B1*, *CTAG1A*, *SCP2D1*, *TKTL2*, and *ACTRT1* before and after TSA treatment in Caco-2 and HCT116 cells. The expression levels were normalized to *GAPDH* mRNA levels. The error bars represent the standard error of the mean for three replicates for each gene. * *p* ≤ 0.05, ** *p* ≤ 0.01.

**Figure 5 pharmaceuticals-15-01319-f005:**
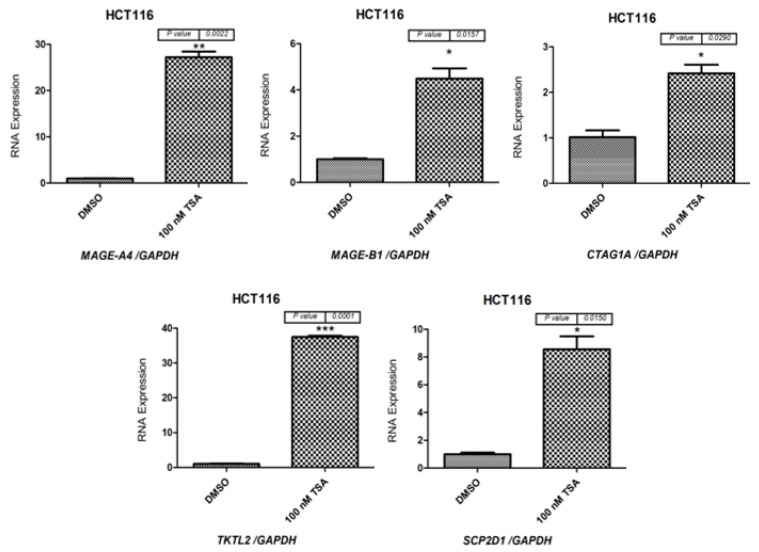
qRT-PCR analysis of *MAGE-A4*, *MAGE-B1*, *CTAG1A*, *TKTL2* and *SCP2D1* expression in HCT116 cells following treatment with 100 nM TSA for 48 h. The bar charts illustrate the expression of *MAGE-A4*, *MAGE-B1*, *CTAG1A*, *TKTL2* and *SCP2D1* before and after TSA treatment in Caco-2 and HCT116 cells. The expression levels were normalized to *GAPDH* mRNA levels. The error bars represent the standard error of the mean for three replicates for each gene. * *p* ≤ 0.05, ** *p* ≤ 0.01, *** *p* ≤ 0.001.

**Figure 6 pharmaceuticals-15-01319-f006:**
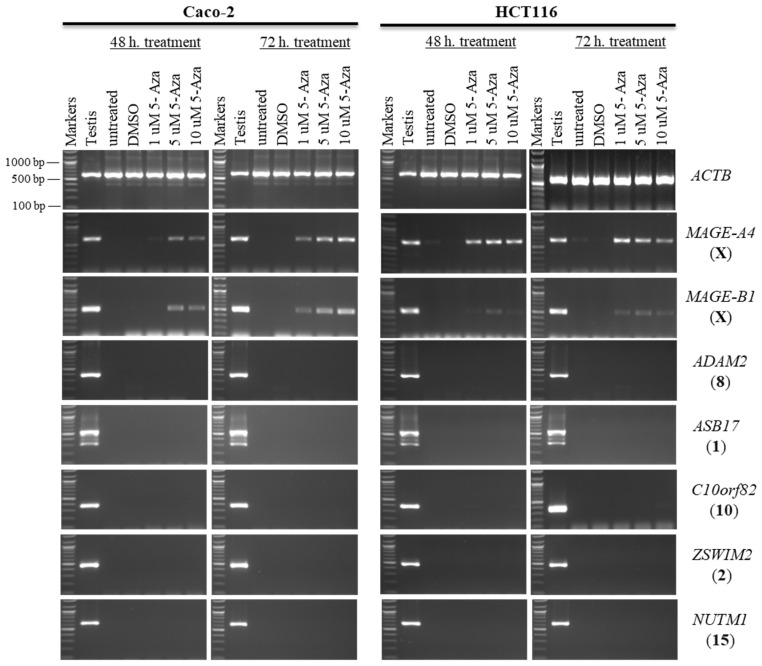
The effects of 5-aza-CdR treatments on testis-specific gene expression profiles in the Caco-2 and HCT116 cancer cell lines. The expression of *ADAM2*, *ASB17*, *C10orf82*, *ZSWIM2*, and *NUTM1* genes is shown on agarose gels following treatment with various doses of 5-aza-CdR for 48 h (left column of each cell) or 72 h (right column of each cell). Untreated Caco-2 and HCT116 cells were used as controls to compare the expression of CT genes with treated cells, and a testis sample served as a positive control for primer efficiency. The control Caco-2 and HCT116 cells were treated with DMSO, as DMSO was the solvent used in the 5-aza-CdR solution. The positive control for the cDNA samples is the expression of the *ACTB* gene. The official names of the genes are written to the right of the agarose gel images, and the location of each gene on the chromosomal is written in parentheses on the right. Above each lane, the particular concentration of 5-aza-CdR (1, 5, and 10 µM) is written.

**Figure 7 pharmaceuticals-15-01319-f007:**
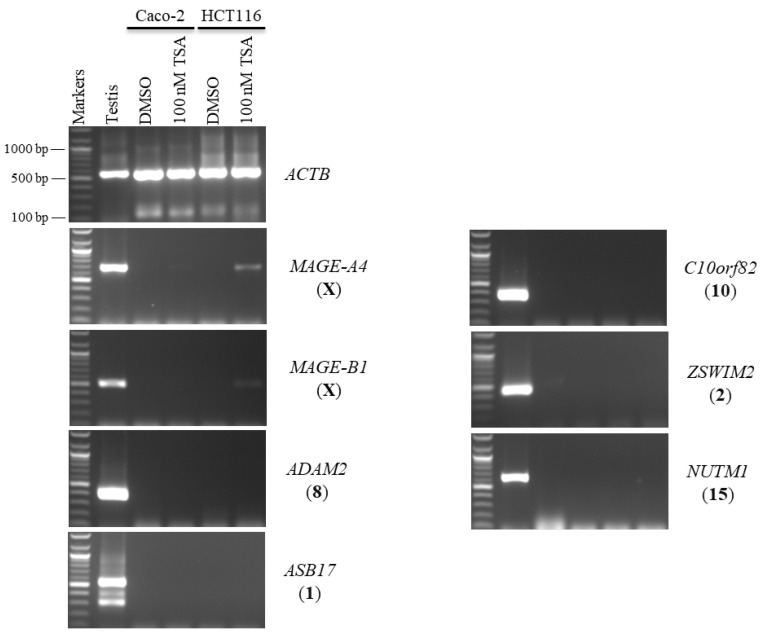
The effects of TSA treatments on testis-specific gene expression profiles in the Caco-2 and HCT116 cancer cell lines. The expression of *ADAM2*, *ASB17*, *C10orf82*, *ZSWIM2*, and *NUTM1* genes is shown on agarose gels following treatment with 100 nM TSA for 48 h. The testis sample served as a positive control for primer efficiency. Control Caco-2 and HCT116 cells were treated with DMSO, as DMSO was the solvent utilized in the TSA solution. The positive control for the cDNA samples is the expression of the *ACTB* gene. The official names of the genes are written to the right of the agarose gel images, and the location of each gene on the chromosomal is written in parentheses on the right.

**Figure 8 pharmaceuticals-15-01319-f008:**
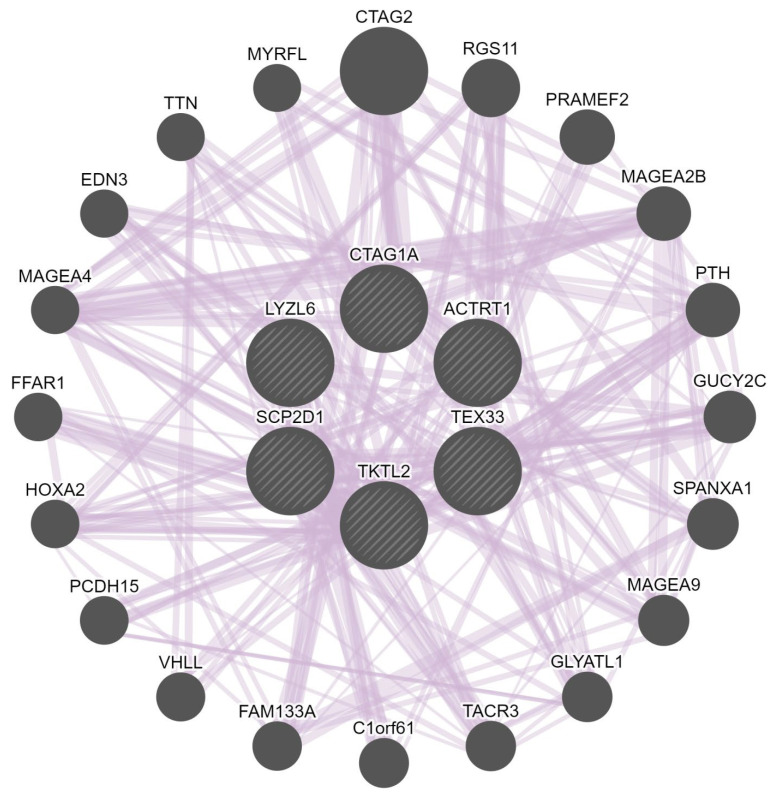
The gene–gene interaction network for the CT genes (SCP2D1, CTAG1A, LYZL6, TKTL2, ACTRT1, and TEX33) generated using the GeneMANIA database. Each central node represents a CT gene (shown as circles). The 20 most frequently neighboring genes found to be strongly linked to the CT genes are shown. The interactions between the CT genes and the other genes are shown as edges (shown as lines).

**Table 1 pharmaceuticals-15-01319-t001:** Sequences of the primers used in the RT-PCR and their estimated product sizes.

Official Gene		Forward and Reverse Primer Directions and Sequences (from 5′→3′)	Ta	Product Size (bp)
Symbol	Full name			
*ACTB*	Actin beta	FP: AGAAAATCTGGCACCACACC	58	553
		RP: AGGAAGGAAGGCTGGAAGAG		
*MAGE-A4*	MAGE family member A4	FP: CTACCATCAGCTTCACTTGC		647
		RP: CTCCAGGACTTTCACATAGC		
*MAGE-B1*	MAGE family member B1	FP: CAGGAATGCTGATGCACTTC		524
		RP: GAGGACTTTCATCTTGGTGG		
*ACTRT1*	Actin related protein T1	FP: GGGATGACATGGAGAAACTC		591
		RP: CCATTTTTGAGAGTCCTGGG		
*ASB17*	Ankyrin repeat and SOCS box containing 17	FP: GTGGGGATATCACTGTTACG		542
		RP: GCACTCTGGAACATAGTACC		
*LYZL6*	Lysozyme like 6	FP: GGCGCTACTCATCTATTTGG		348
		RP: CCGGACACAATCCTTTTTGC		
*TKTL2*	Transketolase like 2	FP: AGGTACTGCATGTGGAATGG		896
		RP: CATCTTCTCCAGTGGATACC		
*ZSWIM2*	Zinc finger SWIM-type containing 2	FP: GACAAACACCTTGGGATTCC		469
		RP: GGCATGAATTGCACTTGTGG		
*ADAM2*	ADAM metallopeptidase domain 2	FP: GTCTTGTTTCTGCTCAGCGG	60	397
		RP: AGCCAACTGAAGACTCCAGG		
*C10orf82*	Chromosome 10 open reading frame 82	FP: CTGCCAAGGAATGTCCAAG		370
		RP: ATGTGCCTTCTTGGCCCTCT		
*TEX33*	Testis expressed 33	FP: GATCCTCCTCGAGAGAGAAC		426
		RP: GCCAGTGTTCTAAGTCCCTC		
*SCP2D1*	SCP2 sterol binding domain containing 1	FP: CAGTTCGAGGTTCTGGGTTC		369
		RP: GCTAAGCAGAACCTTGCCAC		
*NUTM1*	NUT midline carcinoma family member 1	FP: CACCACCAGTTGCTCAACTG		623
		RP: CTCCTTCACAGCTTCTGGTG		
*CTAG1A*	Cancer/testis antigen 1A	FP: CCTGCTTGAGTTCTACCTCG		235
		RP: CTGCGTGATCCACATCAACA		

Abbreviations: FP: Forward primer, RP: Reverse primer, Ta: Annealing temperature, bp: base pair.

**Table 2 pharmaceuticals-15-01319-t002:** Sequences of the primers used in the qRT-PCR and their estimated product sizes.

Official Gene Symbol	Forward and Reverse Primer Directions and Sequences (from 5′→3′)	Product Size (bp)
*GAPDH*	FP: GGGAAGCTTGTCATCAATGG	173
	RP: GAGATGATGACCCTTTTGGC	
*MAGE-A4*	FP: CTACCATCAGCTTCACTTGC	134
	RP: AGCCAACTCATCCACCTTGT	
*MAGE-B1*	FP: GAAGGCAGATATGCTGAAGG	125
	RP: CACTAGGGTTGTCTTCCTTC	
*TKTL2*	FP: CATGGTAAGTGTGGCACTAG	149
	RP: CACAGTGGGAACCAATAAGG	
*SCP2D1*	FP: CCAGCAGACACTGTCTTTAC	129
	RP: CTTCCAGCTAAGCAGAACCT	
*ACTRT1*	FP: GGGATGACATGGAGAAACTC	153
	RP: GCACACTGAAGGTCTCAAAC	
*CTAG1A*	FP: CCTGCTTGAGTTCTACCTCG	132
	RP: CCGGACACAGTGAACTCCTT	

Abbreviations: FP: Forward primer, RP: Reverse primer, bp: base pair.

## Data Availability

Data is contained within the article.

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
