# Peer review of "The Expression Patterns of Human Cancer-Testis Genes Are Induced through Epigenetic Drugs in Colon Cancer Cells"

_pharmaceuticals, 2022, doi:10.3390/ph15111319_

Round 1

Reviewer 1 Report

1. The artionale of the study is very good but it should be written in short and focused manner to common readers.

2. I can see lot of grammar/english eerrors in the manuscript and it is advised to fix with native english speaker.

3. The some of the cited references in introduction are not appropriate to the given content. Please fix it with correct references in introduction.

4.  It iadvised to mention cat# for all consumables/reagens

5.  Mention PCR conditions

6. 2.7, 2.8 can be merged with 2.4 &2.5

7.  what is the rationale behind different Aza concentrations selection, For eg, Figure 1

8. what is the basis for time intervals for eg: 48 h and 72 h, for eg: figure 1

9. Testis and untreated what is the purpose, Figure 1

10. I want to see the raw files of real time PCR

Reviewer 2 Report

The authors in this paper evaluate the upregolation of CT genes in colon cancer cell lines treated with DNMTi and HDACi, treatments that can be useful in the therapies of this type of cancer.

The paper is interesting and well written but I have some question for the authors.

Major point:

as the results obtained with RT-PCR are partially different from those obtained with the agarose gel it would be necessary evaluating all the genes considered in the agarose gel with RT-PCR.

line 218-219 'However, TKTL2 and ACTRT1 transcriptions were activated in HCT116 cells and in both cell lines, respectively'. This results is not evident in Figure 3, there is a mistake.

Minor point:

line 103: correct the apix in the number of cells.

in figure 1: why SCP2D1 is upregulate only with medium doses of 5-AZA. Can be an interesting explanation? the results obtained with RT-PCR can be different.

Round 2

Reviewer 2 Report

the authors made the correction.